# Single-dose testosterone administration increases men's preference for status goods

G. Nave[1], A. Nadler [ID] [2], D. Dubois[3], D. Zava[4], C. Camerer[5] & H. Plassmann [ID] [3,6]

In modern human cultures where social hierarchies are ubiquitous, people typically signal their hierarchical position through consumption of positional goods—goods that convey one's social position, such as luxury products. Building on animal research and early correlational human studies linking the sex steroid hormone testosterone with hierarchical social interactions, we investigate the influence of testosterone on men's preferences for positional goods. Using a placebo-controlled experiment ($N = 243$) to measure individuals' desire for status brands and products, we find that administering testosterone increases men's preference for status brands, compared to brands of similar perceived quality but lower perceived status. Furthermore, testosterone increases positive attitudes toward positional goods when they are described as status-enhancing, but not when they are described as power-enhancing or high in quality. Our results provide novel causal evidence for the biological roots of men's preferences for status, bridging decades of animal behavioral studies with contemporary consumer research.

[1] Marketing Department, The Wharton School of the University of Pennsylvania, 3730 Walnut St., JMHH #700, Philadelphia, PA 19104, USA. [2] Finance Department, Ivey Business School, Western University, 1255 Western Rd., London, ON N6G 0N1, Canada. [3] Marketing Area, INSEAD, Boulevard de Constance, 77300 Fontainebleau, France. [4] ZRT Laboratory, 8605 SW Creekside Pl., Beaverton, OR 97008, USA. [5] Humanities and Social Sciences Division, California Institute of Technology, 1200 E California Blvd MC 228-77, Pasadena, CA 91125, USA. [6] Social and Affective Neuroscience (SAN) Team, Institut du Cerveau et de la Moelle épinière (ICM), Sorbonne Université, INSERM UMR 1127, CNRS UMR 7225, 75013 Paris, France. These authors contributed equally: G. Nave, A. Nadler.  Correspondence and requests for materials should be addressed to G.N. (email: gnave@wharton.upenn.edu) or to H.P. (email: hilke.plassmann@insead.edu)

From schools of fish to modern human communities, social hierarchies are ubiquitous across species[1,2]. Hierarchies give rise to advantages at the group level, such as facilitating leader–follower coordination and reducing resource conflict[3]. At the individual level, higher social rank improves mating opportunities, promotes access to resources, reduces stress, and increases social influence[4–7]. Therefore, individuals exert considerable effort to enhance their social rank by gaining status (i.e., respect and admiration from others, sometimes also referred to as prestige) and power (i.e., control over valuable resources, sometimes also referred to as dominance)[8,9].

How do people achieve higher status? In early human societies, displays of hunting skills and physical aggression were primary in promoting one's standing in society. In contemporary settings, however, hunting and aggression have been replaced by different strategies, such as displays of culturally valued skills and behaviors (e.g., obtaining academic degrees). Another prevalent route to higher status rests on the display of wealth through positional consumption[10,11]. This idea was introduced by Thorstein Veblen's seminal work, *The Theory of the Leisure Class*[12], which describes how wasteful expenditures on positional goods, which display one's apparent resources to others, shape the social strata over time[8]. Such goods are particularly effective signals of status because they separate the "haves" from the "have nots" through economic (e.g., high price) or physical (e.g., restricted access for private club members) barriers. Although Veblen's insights were overlooked by classical market theories, modern economic theories began to incorporate this view by showing that a balance of prices and goods sustains the market for costly signals[13,14]. Indeed, goods that wealthier individuals gravitate toward (hereafter, "positional goods") also tend to be more visible to others than other goods that are more affordable and thus accessible to everyone[15,16].

Understanding the drivers of costly signaling through positional consumption is important because this behavior is, by definition, wasteful—in the sense that less expensive goods could have the same functional value as their high-status counterparts (e.g., cars and houses). Status consumption therefore creates inefficiencies. Spending resources to elevate perceived status might, for instance, perpetuate poverty by reducing self-investment in health and education among the poor, who spend disproportionately more on status signals and thus substitute status signaled through consumption for long-run wealth accumulation[17–19]. While recent work has explored the socio-psychological antecedents of status-driven consumption[20–22], little is known about its biological basis, via genes, hormones, or brain activity.

An analogy to human conspicuous consumption in animal behavior lies in the "handicap principle" of the evolutionary theory of sexual selection[23]. Many species undergo adaptations that wastefully consume physiological resources without yielding immediate survival benefits, such as the stag's heavy antlers and the peacock's vivid train. The handicap principle explains these adaptations as costly signals of male fitness: because only the fittest can afford to waste resources on traits that do not directly increase survival probability, these adaptations become reliable indicators of fitness. Moreover, given that the proximal purpose of such adaptations is to promote the spread of genes by increasing attractiveness to mates, these traits must be displayed conspicuously—hence the length of the stag's antlers and iridescence of the peacock's tail.

The male sex steroid hormone testosterone (T) is associated with a range of male reproductive and social behaviors in non-human and human species. In non-humans, individual differences in T levels have been linked to social rank[24,25], and a context-sensitive rise of T during the breeding season is associated with conspicuous displays of costly signals, such as complex courtship singing in male birds and the growth of bulky antlers in stags[26–28].

In humans, too, T levels can situationally increase in contexts related to social rank and male reproductive behavior[29], e.g., during competitions and after winning them, in the presence of an attractive mate, and even following acts of conspicuous consumption, such as driving a sports car (vs. a family sedan)[30]. While early human studies (conducted mainly among prisoners) reported correlations between T and aggression[31], subsequent research has proposed that T does not increase aggression per se[32], but rather the motivation to promote one's status[24,33,34]. These studies (conducted in both males and females) showed that pharmacologically elevated T increased generosity[35], cooperation[36], and honesty[37], all of which are pro-social non-aggressive behaviors that may promote one's status[24]. Other studies further reported intriguing correlations between the 2D:4D digit ratio, a candidate proxy of prenatal testosterone exposure (though see ref. [38]), and behavioral measures of courtship-related consumption[39,40] (a relation that was not evident in our own data).

Building on Veblen's theory of positional consumption, as well as the evidence that a situational increase in T leads to rank-promoting behaviors in animals and humans, we hypothesized that elevated T levels would cause men to exhibit stronger preference toward goods that promote their social rank. To test this hypothesis, we randomly administered either T ($N = 125$) or placebo ($N = 118$) topical gel to 243 males, following a double-blind administration protocol[41,42]. The sample size was maximized according to the study's budget constraints.

Participants completed two tasks. In the first task, we showed participants pairs of apparel brands that differed in their associations with social rank and asked them to indicate their preferences for one or the other. The second task investigated whether T influenced attitudes toward the same goods when they were positioned differently. Specifically, we measured participants' attitudes toward products that were positioned either as status-enhancing, power-enhancing, or high in quality.

The results confirmed our hypothesis: we found that participants who received T showed greater preference for brands that were associated with higher social rank (task 1), and that T increased positive attitudes toward goods that were positioned as status-enhancing but not those positioned as power-enhancing or high in quality (task 2). We thus conclude that T elevates men's desire to promote their social status through economic consumption.

## Results

**Manipulation check.** To monitor the levels of T and other hormones that might influence decision making during the experiment (e.g., cortisol)[43], participants provided one pre-treatment saliva sample (that also allowed us to investigate the correlation between behavior and basal T) and three post-treatment saliva samples, that were assayed by liquid chromatography tandem mass spectrometry (LC-MS/MS). We observed elevated post-treatment saliva T measurements in the T group compared to the placebo group, providing a robust manipulation check (see Fig. 1). There were no treatment effects on the levels of hormones that were not expected to be influenced by it (Supplementary Table 3) or mood (Supplementary Table 4)

**Testosterone's effect on brand preference.** Participants viewed five pairs of pretested apparel brands in a randomized, counterbalanced order. One brand of each pair was associated with higher social rank than the other (e.g., Calvin Klein, high vs. Levi's, low). Importantly, perceived social rank difference between

the brands in each pair was substantially greater than perceived difference in quality, mitigating the possibility that the latter influenced participants' preferences in our task (see Fig. 2b and Supplementary Table 5). Participants indicated the extent to which they preferred one brand relative to the other using 10-point rating scales (Fig. 2a).

A series of mixed-effects linear models that included random intercepts for participant and brand pair examined the effect of T

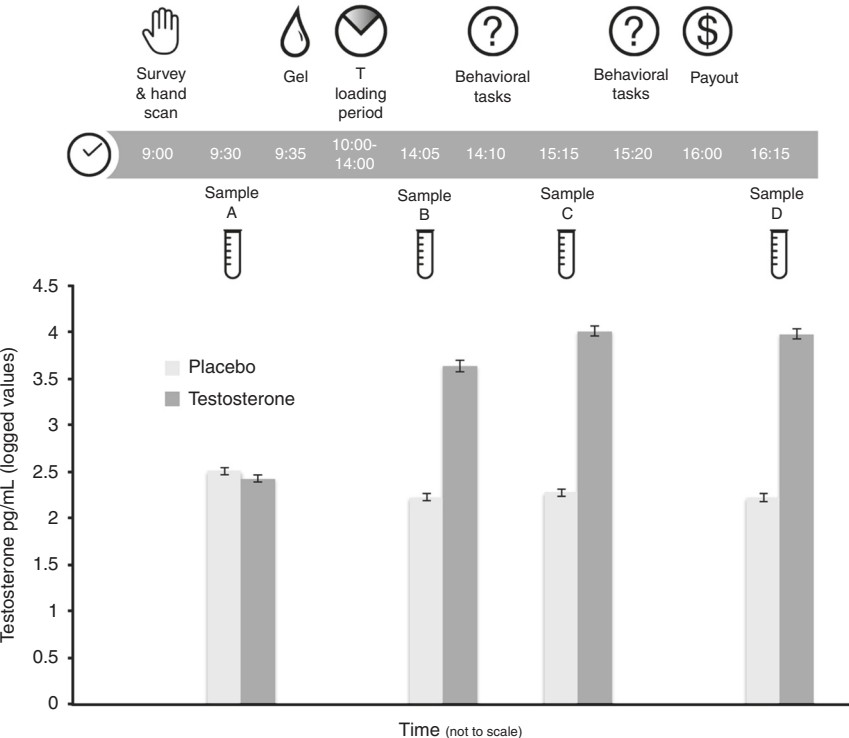

**Fig. 1** Experimental setup and salivary testosterone levels. Participants ($N = 243$) arrived at the lab at 9:00 a.m., had their hands scanned to take 2D:4D measures, completed an intake survey, and gave a baseline saliva sample ("A") before application of either T or placebo topical gel. After a 4-h loading period, participants came back to the lab and took part in a battery of behavioral tasks. Three additional saliva samples ("B," "C," and "D") were collected during the experiment, all of which indicated elevated T levels in the treatment group compared to the placebo group. The behavioral tasks reported were the main focus of the study, and took place immediately following the participants' return the lab in the afternoon (after saliva sample B). Error bars denote s.e.m.

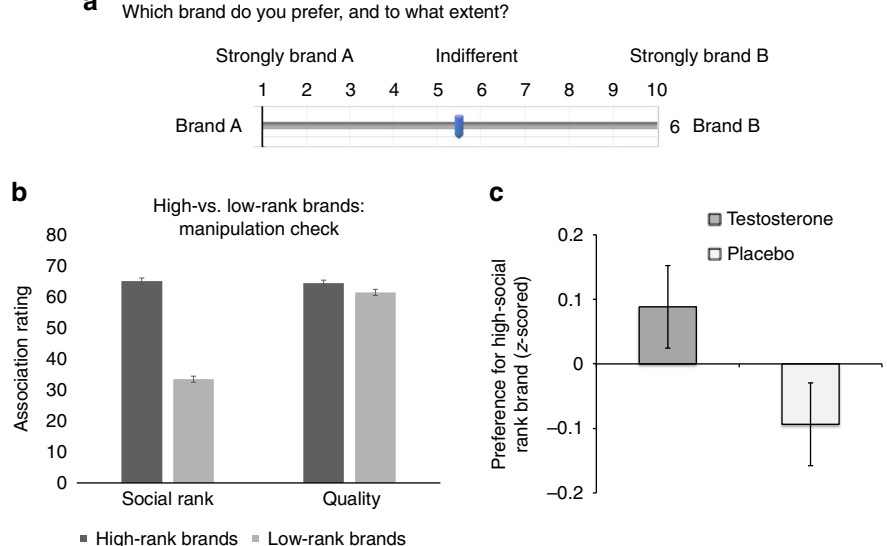

**Fig. 2** Task to assess preference for brands high versus low in social rank and results. **a** Preference task showing the setup and main dependent variable. **b** Mean social rank and quality association ratings of brands pre-classified based on a pretest as high vs. low rank, by main study participants ($N = 243$); perceived differences in social rank associations between the brands were substantially greater than the difference in perceived quality. **c** Mean preference toward the high (versus low) social rank brands for the two treatment groups (z-scored at the pair level). Error bars denote s.e.m. For the corresponding dot plot, see Supplementary Fig. 2

on brand preferences (with higher ratings indicating greater preference for the high over the low social-rank brands). We found that the T group preferred the brands with higher social rank compared to the placebo group (standardized $\beta = 0.18$, 95% CI = [0.045 0.359], $z = 2.01$, $p = 0.04$; Fig. 2c and Supplementary Table 7). The effect was robust when controlling for age, mood, treatment expectancy, the 2D:4D digit finger ratio (a proxy of prenatal T exposure), and post-treatment levels of all measured hormones that were unaffected by T treatment.

Additional analyses revealed that participants' baseline (pre-treatment) T levels were also associated with greater preference for brands with higher social rank ($\beta = 0.13$, 95% CI = [0.042 0.220], $z = 2.867$, $p = 0.004$; see Supplementary Table 12; the treatment effects were robust to controlling for baseline T), suggesting that the effect of T is both "activational" (i.e., behavior is influenced by transient changes in T levels) and "dispositional" (i.e., behavior is related to baseline T levels, which are relatively stable when measured within individuals at the same time of day). We found a similar effect for baseline levels of androstenedione ($z = 2.53$, $p < 0.01$), a downstream metabolite of testosterone. Levels of androstenedione correlate with testosterone (in our sample $r = 0.44$, $p < 0.01$), and it has been suggested (by animal studies) to have potent behavioral effects that are similar to those of T (see Supplementary Table 12).

Finally, we investigated whether the shift toward brands high in social rank was driven by a shift in their rank-related associations, rather than a preference shift per se. We found that the social rank associations of the brands used in our task were not affected by T treatment (see Supplementary Table 6).

**Testosterone's effect on product liking.** The second task investigated the effect of T on two distinct paths individuals can use consumption to climb the social ladder: increasing status (defined as the prestige, respect, and admiration an individual has in the eyes of others) or power (defined as feelings of control over valued resources)[8,9]. Although power and status are inextricably intertwined in most animal social groups, the two can be decoupled within human social contexts. For example, a political adviser unknown to the public can have significant control over important decisions without receiving social recognition (high power, low status); conversely, a well-known academic may enjoy high status and be respected by the public but have little power over policy decisions regarding her research findings[8]. This experiment provides a unique opportunity to disentangle the extent to which T affects power- versus status-enhancement motives. (Note that apart from previous finding suggesting a relation between T and status seeking, there is also evidence that circulating T levels are associated with implicit power motivation.)[44,45]

Participants viewed brief text descriptions of six different goods in a randomized, counterbalanced order and then indicated their attitudes toward each using three 10-point scales (favorable/unfavorable, good/bad, positive/negative). For each good, we composed and pretested three different text descriptions, identical except for specific phrases emphasizing associations with status, power, or high quality (Fig. 3a). Descriptions were randomized so that each participant saw two goods in each condition (status, power, quality) in a counterbalanced fashion. This yielded a 2 (treatment: testosterone, placebo) × 3 (association: status, power, quality) between-participants factorial design, repeated within-participants over six different goods (see Fig. 3 and Supplementary Methods). Participants also indicated their hypothetical purchase intentions (10-point scale) and willingness to pay (WTP, open text entry) for the goods; these measures were standardized and averaged to create an index for hypothetical purchasing behavior.

A series of mixed-effects linear models that included product and participant random intercepts and participant random slopes for positioning conditions estimated the effect of T on product attitudes. We found that the T group had more positive attitudes toward goods described as status-enhancing compared to the placebo group (reflected by a significant treatment × status interaction, standardized $\beta = 0.275$, 95% CI = [0.042 0.508], $z = 2.31$, $p = 0.02$; Fig. 3c and Supplementary Tables 8–9). However, there was no difference between the two treatment

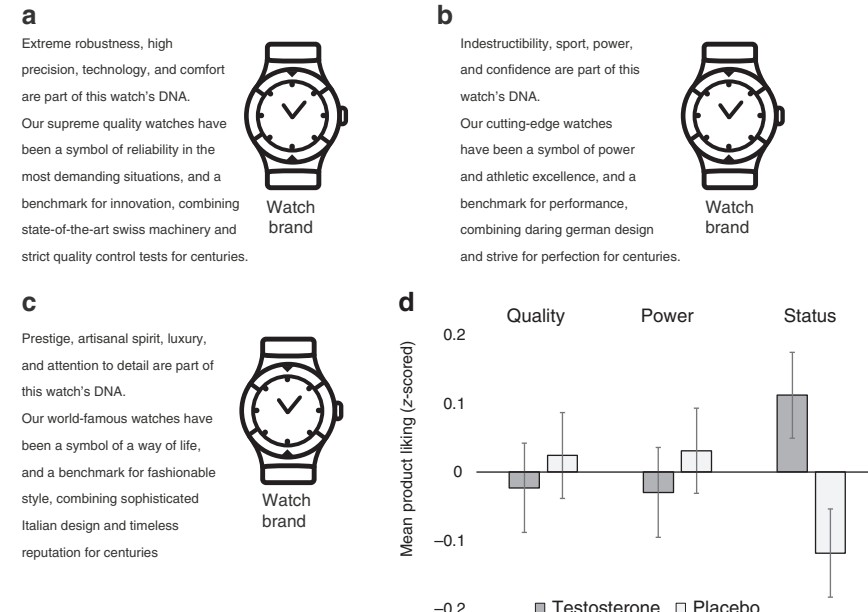

**Fig. 3** Task measuring attitudes towards identical goods associated with different rank-enhancing strategies and results. For each of six goods we created three different text ads emphasizing either its associations with (**a**) high quality, (**b**) power, or (**c**) status. The ads were identical otherwise. **d** Mean attitudes of the goods ($N = 243$ participants) for each of the three conditions (z-scored at the ad level). Error bars denote s.e.m. For the corresponding dot plot, see Supplementary Fig. 3. The watch clipart was created by Professor Amos Nadler

groups when the same goods were described as power-enhancing ($\beta = -0.016$, 95% CI $= -0.242\ 0.209]$, $z = -0.14$, $p > 0.89$) or high in quality ($\beta = -0.044$, 95% CI $= [-0.243\ 0.155]$, $z = -0.44$, $p > 0.66$). Results were robust to controlling for age, mood, treatment expectancy, 2D:4D, and hormones that were unaffected by T treatment. The results hold when using either quality or power as the baseline category, indicating that T's positive effect on preferences for status was significantly greater than its effects on preferences for either quality or power (see Supplementary Table 11).

An analogous analysis for the secondary measure of hypothetical purchasing behavior revealed a similar yet statistically weaker pattern (see Supplementary Table 10). We found a marginally significant treatment × status interaction ($\beta = 0.160$, 95% CI $= [-0.023\ 0.342]$, $z = 17$, $p < 0.10$). There were neither treatment × power ($\beta = 0.02$, 95% CI $= [-0174\ 0.215]$, $z = 0.36$, $p > 0.80$) nor treatment × quality ($\beta = -0.042$, 95% CI $= [-0.206\ 0.123]$, $z = -0.495$, $p > 0.62$) interactions for the intentions measures.

As in task 1, we performed additional analyses exploring the association between-participants' baseline (pre-treatment) T and their attitudes toward the goods. However, we did not detect a reliable baseline T × status interaction, though the coefficient was positive in sign (see Supplementary Table 13; the treatment effects were robust to controlling for baseline T).

## Discussion

Taken together, these findings suggest that the consumption of positional goods may stem, at least partly, from biological motives. By adopting an evolutionary perspective, we contribute to a growing body of work in economics uncovering the adaptive function of consumption and complement the increasingly rich nomological network around how status processes govern individuals' thoughts, feelings, and behaviors[46].

Our results demonstrate for the first time that T causally influences rank-related consumer preferences (task 1), and that the effect is driven by status enhancement and not power motives or inclination for high quality (task 2). These findings contribute to a burgeoning literature shedding light on the association between T and humans' desire for status. Of note, the bulk of efforts to date mostly relied on behavior in economic games, in which participants make trade-offs between monetary and social rewards. As both reward types can promote one's status, it is not always clear, a priori, what behavior should increase status in such games. For example, one study found that T administration increased cooperation;[36] a different study[47] found that T reduced cooperation. In both cases, the authors postulated that T-induced status-seeking behavior caused the effect, by proposing opposite influences of status-seeking behavior on cooperation. The current study overcomes this limitation by directly measuring and manipulating status preferences.

Our findings may be useful for generating new hypotheses regarding contexts in which positional consumption occurs[17–19]. Men experience situational elevation in T during and following sporting events, in the presence of attractive mates, and following meaningful life events such as graduation and divorce[29,48,49]. Our results suggest that in such contexts, male consumers might be more likely to engage in positional consumption, and might find status-related brand communications more appealing. We hope that our findings will guide further research exploring how contextual variations in real-world settings that change T levels (e.g., sporting events, changes in marital status) affect status-related economic preferences.

The current work has several limitations, which represent opportunities for future research. First, our behavioral measures did not involve actual purchases. Although the pattern of results for our secondary, non-incentivized index of hypothetical purchase behavior in task 2 was consistent with the primary results of T's effect on attitudes toward positional goods, the effects were smaller in magnitude (statistically significant only at the $p < 0.10$ level, and even weaker when taking into account various controls; see Supplementary Table 10). However, attitudes toward the target goods predicted both hypothetical purchase intentions ($r$ $(1,446) = 0.329$, 95% CI $= [0.282\ 0.374]$, $p < 0.001$) and WTP ($r$ $(1,441) = 0.249$, 95% CI $= [0.200\ 0.297]$, $p < 0.001$), suggesting a link between these measures. Of note, research suggests that hypothetical purchase intentions and WTP do not always reflect precise value signals on a reported[50] or neural[51] level. Furthermore, unlike attitudes, purchase intentions and WTP might be more subject to factors tied to unmeasured individual differences (e.g., budget constraints). Thus, these measures might have been too noisy to detect a statistically significant effect in our study. Future research should further investigate T's influence on incentive-compatible economic behaviors promoting social rank, controlling for individual differences as best as possible.

Second, to increase the ecological validity of our pharmacological treatment, we administered T using a widely prescribed gel at a typical daily dose (100 mg) that leads to elevated serum T levels within the normal male physiological range[41]. This design choice might have led to effects that are moderate in size (task 1: $d = 0.18$; task 2: $d = 0.28$), though sufficiently large for detection in our study. Future research should explore whether dose, timing of exposure (i.e., the lag between T administration and the behavioral task), and delivery type (e.g., topical, oral) might lead to different effects on behavior.

Third, the causal connection between T and status appears to be bi-directional. We find that changes in T causally promote attitudes toward status-enhancing goods, and other studies have shown that consumption of such goods changes T levels[30]. This relationship suggests a T-mediated process in which status consumption and T level might reinforce each other[8], and calls for further investigations of mutual influence of T and status consumption over time. Moreover, given that a reliable association between baseline T levels and consumer preferences emerged only in the first task, and that this is the first study testing this relationship, more research is needed to establish robustness and variation of a dispositional effect of T on consumer preferences.

Furthermore, while the effects reported here were robust when controlling for mood, beliefs about treatment, and levels of other hormones that might have influenced behavior, we cannot entirely rule out that the effects of T on economic behavior were generated by other indirect mechanisms. For example, we could not fully control participants' behaviors during the drug loading period (i.e., the time interval between when the T gel was applied and when they returned to the lab).

Fourth, because the T system is sexually dimorphic, and given that most of the behavioral literature in animals is concerned with males, we relied on an all-male sample (the use of same-sex participants is a common practice in the literature). It is important to note, however, that women also engage in conspicuous consumption, and preliminary evidence suggests that biological factors (including hormones that relate to the menstrual cycle) are involved[52–54]. As there is evidence that T promotes status-related behaviors in females[24,35,36], further research should explore whether the effects of T on consumer preferences are generalizable to females, while taking into account that which brands and goods are status-enhancing is likely to differ across sexes[55].

Finally, it is important to keep in mind that cultural differences might play a role in the biological underpinnings of status behavior and that status signals are not universal (e.g., status of

different occupations varies substantially across countries[56]). Preliminary evidence suggests that individual cultural differences such as self-construal may moderate the behavioral effects of T[57]. Moreover, some cultures frown upon overt expression of material status. Replicating our results in different cultures would provide an opportunity to test whether the expression of status is uniquely culturally concordant or catholically materialistic. Overall, future research should aim to generalize our findings to other populations to create robust empirical foundations for the biological basis of consumer preferences.

## Methods

**General procedure**. Two hundred forty-three males, aged 18–55 (M = 23.63, SD = 7.22), mostly (217, 89%) students at a private Southern California consortium, participated in the study. Non-student participants were community members from surrounding cities. For pre-screening criteria see Supplementary Methods; detailed demographic characteristics of the two treatment groups are available on Supplementary Table 1. The institutional review boards of Caltech and Claremont Graduate University approved the study, all participants gave informed consent, and no adverse events occurred.

Similar to a previously described study (42), participants arrived at the lab at 9 a.m., signed informed consent forms, and proceeded to a designated room for hand scanning (see Supplementary Methods). Participants were then randomly assigned to private cubicles, where they completed demographic and mood questionnaires (see Supplementary Methods) and provided an initial saliva sample by passive drool. Next, participants proceeded to gel application (further details below), after which they were instructed to refrain from bathing, any activity that might cause excessive perspiration, and direct physical contact with females before the afternoon session; finish eating no later than 1 p.m.; and return to the lab promptly at 1:55 p.m. well hydrated. Participants were given printed material containing these precautions and instructions prior to dismissal.

Participants returned to the lab at 2 p.m. (no participant was late), provided a second saliva sample, and began the behavioral experiment in the same cubicle they had occupied in the morning. The experiment lasted approximately 2 h and consisted of a battery of behavioral tasks, none of which included feedback about monetary payoffs or performance. Only the final task included feedback regarding the participants' performance relative to others. The rationale for conducting a battery of tasks is maximizing the knowledge gained from each human participant undergoing a pharmacological manipulation, a practice that is standard[58,59]. The two tasks reported in the paper were focal and therefore were conducted at the outset, immediately after participants' arrival at the lab in the afternoon and the first post-treatment saliva sample. On average, participants earned $68.12 USD (SD = $17.36) for participating in the experiment. Payout varied as a function of their performance in some of the other tasks.

To standardize hormonal measurements among participants, we did not randomize the order of the behavioral tasks, in similar fashion to previous studies[58,59]. Following the experiment, participants completed an exit survey, where they indicated their beliefs about the treatment they had received, using a five-point scale, and were privately paid in cash.

**Testosterone administration**. Participants were escorted in groups of two to six to a semiprivate room where a research assistant provided a small plastic cup containing 10 g of clear gel and stated that it was equally likely to contain T or placebo. The cups were filled in advance by the lab manager, who did not interact with participants and did not reveal the contents of the cup to the assistant. The gel contained either topical T 1% (2 × 50 mg packets Vogelxo® by Upsher-Smith) or the volume equivalent of an inert placebo (80% alcogel, 20% Versagel®). Participants were instructed to remove the clothing from their upper bodies and apply the entire contents of the gel container to their shoulders, upper arms, and chest, as demonstrated by the research assistant, and were told to wait until the gel fully dried before putting their clothes back on.

We chose to administer T using topical gel as this was the only T administration method for which the pharmacokinetics of a single-dose administration had been investigated at the time[41]. That study[41] demonstrated that plasma T levels peaked 3 h after single-dose exogenous topical administration, and that T measurements stabilized at high levels during the time window between 4 and 7 h following administration. Therefore, we had all participants return to the lab 4.5 h after receiving gel, when androgen levels were elevated and stable.

**Saliva sampling**. Each participant provided four saliva samples by passively drooling into a plastic tube, at predetermined sampling times throughout the study: (1) before treatment administration; (2) upon return to the lab, just prior to starting the behavioral tasks; (3) in the middle of the behavioral tasks battery; and (4) following the one and only task involving performance feedback, at the end of the experiment. We used saliva samples to avoid potential stress that might be induced by high-resolution blood drawing throughout the experimental session. Each saliva sample was time stamped. No food or drinks were allowed into the

laboratory, and the only water given to the participants was after their third saliva draw (an hour before the fourth and final saliva draw).

To allow robust manipulation checks and obtain control for the participant's biological state, we used LC-MS/MS (detection levels and precision are available in Supplementary Table 2) to measure the following salivary steroids: estrone, estradiol, estriol, testosterone, androstenedione, DHEA, 5-alpha DHT, progesterone, 17OH-progesterone, 11-deoxycortisol, cortisol, cortisone, and corticosterone. A series of one-sample Kolmogorov–Smirnov tests for conformity to a Gaussian distribution (Supplementary Table 3) indicated that all hormonal measurement distributions were best approximated by a Gaussian distribution following a log-transformation, as indicated by higher p-values. Thus, all hormonal measurements were log-transformed prior to data analysis in order to make their distributions closer to Gaussian. We provide further technical details of the procedure and analysis of hormonal changes following the treatment in Supplementary Table 3.

**Task 1: Testosterone's effect on preference for brands high in social rank**. In a pretest, we presented 184 students of a private Southern California college (with similar demographic characteristics as our participants) with the logos of 15 familiar apparel brands in a randomized, counterbalanced order. Participants rated each brand's association with quality and social rank using 100-point subscales (0 = not at all to 100 = very much). Social rank was constructed by averaging three items related to status (status, conspicuousness, prestige[60,61]) and three items related to power (power, performance, control[62–64]).

Importantly, the first task did not allow us to directly disentangle status enhancement and power enhancement motives, as we could not identify (based on the pretest data) any brand pairs that were perceived differently with respect to their status and power associations, and thus we combined the average of the six items to a general measure of social rank associations. Our data indicated that brands high in social rank were typically also perceived as high in quality. However, we were able to identify five pairs of brands for which the difference in social rank associations was significantly greater than the difference in quality associations. Supplementary Table 5 summarizes these pairs, along with their perceived social rank and quality associations among the experiment participants. In order to mitigate the possibility that participants would guess the study's purpose, the task included an additional pair for which both brands were associated with lower social rank (Gap vs. H&M).

In the experimental task, we presented participants with the five brand pairs in a randomized, counterbalanced order (Fig. 2a). One brand appeared on the left side of the screen and the other on the right (sides were randomized and counterbalanced). Participants indicated which of the two brands they preferred using a 10-point Likert rating scale (1 = strongly left brand, 10 = strongly right brand). For standardization, we z-scored the ratings at the question level; all of the results are robust to this analytical choice.

We followed the behavioral task with a survey that examined the participants' associations with the brands used. We showed participants all brands in a randomized order and asked them to rate their associations with quality and social rank (i.e., power and status) using 100-point scales. We constructed a social rank scale by averaging their power and status ratings in a similar fashion to the pretest. This scale allowed us to examine whether T affects preferences for social-rank-enhancing brands rather than forming social rank associations.

Using the post-experimental survey, we conducted a manipulation check verifying that our pairs of brands differed in social rank associations more than they differed in quality associations, for the participants of our main study (Supplementary Table 5). Paired t-tests of the difference in difference between social rank and quality associations showed that the differences in social rank associations were greater than the differences in quality associations for all of the brand pairs (all $p$'s < 0.001).

To rule out the potential effect of T administration on brand associations rather than a difference in preference for these brands, we tested for effects of T on the perceived quality and social rank associations of the brands, using two-sample t-tests (Supplementary Table 6). We found no reliable treatment effects on any of the brands' rank perceptions (all $p$'s > 0.25). Only one of the seven brands showed a significant difference (at the $\alpha = 0.05$ level, uncorrected) in quality perception. Thus, T did not influence the brands' perception among our participants.

**Task 2: Testosterone's effect on attitudes toward goods associated with status, power, or quality**. For each of six goods we composed three different text descriptions, describing the goods as either power-enhancing, status-enhancing, or high in quality. The descriptions included the goods' images accompanied by the text (all ads are available online in the project's open science framework page). We pretested the different text descriptions in two separate online surveys (N = 714 and N = 744 Amazon Mechanical Turk users). Participants saw one of the three descriptions for each of the goods in a counterbalanced randomized order and reported to what extent the descriptions and the goods conveyed status, power, and quality on a 10-point Likert scale (0 = not at all, 10 = very much). As in the pretest for Task 1, respondents rated the descriptions' and goods' associations with quality, three items related to power, and three items related to status. The pretest results are summarized in Supplementary Fig. 1.

In the experimental task, we manipulated social rank (i.e., power and status) and quality associations for identical goods and investigated whether T administration altered attitudes toward these goods. (We included high quality as a third condition to conceptually replicate the findings of study 1 and assess the extent to which social rank-promoting behaviors might stem from a preference for characteristics typically associated with high-end options, such as quality, as opposed to deeper psychological motives directly tied to social rank promotion. This is important because perceived quality is often influenced by price and brand effects.)

We presented each participant with one of the three text descriptions of each good. Each participant saw the text descriptions for all six of the goods, such that two of the descriptions focused on quality, two on power, and two on the status features of the goods. We randomly assigned each participant to one of three groups that saw a different combination of goods × text description interaction (i.e., a third of the participants saw the status description, another third the power description, and another third the quality description for each of the goods, such that two out of the six descriptions were in each description condition for each participant). This resulted in a 2 (T/placebo treatment, between participants) × 3 (description condition, between participants) factorial design repeated over six good categories (within participant).

Participants reported their attitudes toward each good (e.g., "What is your attitude toward Alpina watches?") using three 10-point Likert scales (1 = unfavorable, 10 = favorable; 1 = bad, 10 = good; 1 = negative, 10 = positive) that were averaged to create a single attitude score. The attitude score was $z$-scored at the text description level (all results are robust to this analytical choice). In addition, we asked participants to report hypothetical purchase intentions (10-point Likert scale) and WTP (open text entry); the two measures were $z$-scored and averaged to create an index for hypothetical purchasing behavior. We found that the attitude ratings predicted both hypothetical purchase intentions ($r$ (1,446) = 0.329, 95% CI = [0.282 0.374], $p < 0.001$) and WTP ($r$ (1,441) = 0.249, 95% CI = [0.200 0.297], $p < 0.001$).

To account for two sources of variance in purchase intentions, we included two task-related questions in our post-experiment survey, after completion of the full experimental battery. First, we asked participants whether they already owned goods in the target category. Second, we asked participants about their general buying intentions for the category (i.e., participants were asked "Within the next month, how likely are you to purchase goods of the following categories?" measured on 1–10 Likert scales). Our regression models included controls for these two factors, both of which were highly significant predictors ($p$'s < 0.001) of hypothetical purchasing behavior.

**Data analysis**. Data were analyzed using linear regression mixed models with item-specific and participant-specific random effects[65]. All estimated models and their detailed results across experimental tasks are available in the Supplementary Information online.

Availability of materials and data. Materials, data, and analysis scripts are available on the project's Open Science Framework (OSF) page: https://osf.io/jqmnx.

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

## Acknowledgements

We would like to acknowledge funding for this research project from INSEAD Research and Development funds to H.P. and D.D., the MacArthur Foundation, Ivey Business School, IFREE, the Russell Sage Foundation, the Wharton Neuroscience Initiative and the Wharton-INSEAD alliance. Special thanks to Jorge Barraza, Austin Henderson, and Garrett Thoelen for research assistance.

## Author contributions

G.N., A.N., D.D., D.Z., C.C., and H.P conceived and designed the study and wrote the manuscript. G.N. and A.N. conducted the experiments and analyzed the data. G.N. and A.N contributed equally to this work.

## Additional information

**Competing interests:** The authors declare no competing interests.

