## [Peer Review File · Nature Communications]

Reviewers' comments:

Reviewer #1 (Remarks to the Author):

This paper demonstrates that the exogenous administration of testosterone (via a gel) to a group of men increases their preference for status products (across two different experimental tasks). The paper was clearly written, and the methodology and associated data analyses seemed appropriate. To the extent that this causal link has yet to be empirically demonstrated using an exogenous administration of testosterone, it offers a novel empirical insight. That said, the finding fails on a very important epistemological criterion, as originally set forth by Davis (1971). Regrettably, it is unlikely to trigger an emphatic "That's Interesting!" response across the interdisciplinary readership of *Nature Communications*. Furthermore, the main finding (while novel in a strict empirical sense is not so from a theoretical perspective) does not seem to pass the necessary threshold that would warrant publication. The current paper seems best suited for a flash report (or short report) in a more specialized journal.

I should add that there is an important theoretical issue that was not addressed in this paper. Is the exogenous administration of testosterone meant to mimic (in an ecological sense) the situational and temporary increase in testosterone (e.g., as might occur via an increase in one's social status following a job promotion)? A more dispositional (rather than situational) perspective would posit that since men exhibit individual differences in terms of their baseline circulating testosterone levels, those with higher baseline T levels are more likely to prefer status products. Hence, a possible additional study that would increase the contribution of this work would seek to establish whether or not the dispositional T effect is operative.

Finally, several key references are missing. For example, when discussing Zahavian signaling in a consumer setting, it is imperative that the authors cite Saad (2007), as he offered a detailed analysis of this exact mechanism. Furthermore, when discussing the digit ratio, the authors should cite the numerous recent papers that have been published linking the digit ratio to consumer-related phenomena (cf. Nepomuceno et al., 2016a, 2016b). Others papers worth citing including Nelissen and Meijers (2011) and Stanton (2017).

References:

Davis, M. S. (1971). That's interesting! Towards a phenomenology of sociology and a sociology of phenomenology. *Philosophy of Social Science*, 1, 309–344.

Nepomuceno, M., Saad, G., Stenstrom, E., Mendenhall, Z., & Iglesias, F. (2016a). Testosterone & gift giving: Mating confidence moderates the association between digit ratios (2D:4D and rel2) and erotic gift giving. *Personality and Individual Differences*, 91, 27–30.

Nepomuceno, M., Saad, G., Stenstrom, E., Mendenhall, Z., & Iglesias, F. (2016b). Testosterone at your fingertips: Digit ratios (2D:4D and rel2) as predictors of courtship-related consumption intended to acquire and retain mates. *Journal of Consumer Psychology*, 26, 231–244.

Nelissen, R. M. A., and Meijers, M. H. C. (2011). Social benefits of luxury brands as costly signals of wealth and status. *Evolution and Human Behavior*, 32, 343–355.

Saad, G. (2007). *The Evolutionary Bases of Consumption*. Mahwah, NJ: Lawrence Erlbaum.

Stanton, S. J. (2017). The role of testosterone and estrogen in consumer behavior and social & economic decision making: A review. *Hormones and Behavior*, 92, 155–163.

Reviewer #2 (Remarks to the Author):

In this article, the authors seek to provide causal evidence that exogenous testosterone administration in a sample of men increases their preference for status brands, as well as favorable attitudes toward goods that are indicative of social status compared to those suggestive of either power or quality.

I commend the authors for having undertaken an intricate and laborious experiment lasting several hours. It was fun to read. However, I point to some important limitations. They are included below, in no particular order:

1) The authors mention that we don't know much about the biological underpinnings of status-driven consumption. First, why is this important? This question remains unanswered throughout the paper. Second, when they discuss the connection between status and male testosterone, it is mostly in nonhumans. Related to this point is what Sapolsky (1997) describes as a limitation in our understanding of behavioral endocrinology, namely that we often (and mistakenly) believe that our testosterone is related to behavior; instead, we should ask whether the opposite is at play. As such, I think that the authors should be open to the possibility of reverse causality here when discussing their limitations.

Sapolsky, R.M. (1997). *The trouble with testosterone*. New York, NY: Scribner.

2) In lines 44-49, the authors could have captured Veblen's basic idea more effectively. In his treatise, he criticized contemporary economists for their use of classical market theories to explain a growing and puzzling phenomenon. Using other disparate disciplines (e.g., anthropology), he described that, as societies shifted from hunter gathering to agriculture and more industrialized pursuits, accumulated wealth replaced aggression as a means for men to signal their standing in society. On a related point, Veblen's book was published in 1899; the reference included here should not contain Martha Banta's name.

3) Line 68: "animals and humans..." Humans are animals (Kingdom Animalia)!

4) Lines 71-72: "subsequent research has proposed indirect evidence that T does not increase aggression per se..." One recent study, in fact, found direct evidence that testosterone increases did not lead to various forms of aggression. Here is the reference:

Vongas, J.G., & Al Hajj, R. (2017). The effects of competition and implicit power motive on men's testosterone, emotion recognition, and aggression. *Hormones & Behavior*, 92, 57-71.

5) In line 77, the authors should note that this paper (Saad & Vongas, 2009) does not report any empirical tests of mediation regarding audience effects (i.e., "visibility of the consuming act"). This should be fixed because it is misleading in the current form.

6) Apart from what previous studies have found, what is the theoretical basis for formulating the hypothesis in the current manuscript (lines 79-80)? Papers that are presented by the authors have been guided by a theoretical framework (e.g., Zahavi's handicap principle or costly signaling theory) so readers might be left guessing as to what is driving the hypotheses here.

7) Subject effect: Given that it was not possible to control what Ss did in between getting the testosterone gel application and returning to the lab, and apart from guidelines to avoid contact with females, what other measures did the authors use to protect the study's internal validity? Was there a

manipulation check at the end of the experiment, prior to debriefing, that gauged the extent to which Ss might have guessed the hypotheses being tested here (and to exclude participants on this basis)?

8) Related to the above point #7: How much were Ss paid for participating?

9) Line 147: The authors should remove "exciting." While I agree that future research opportunities are exciting, I feel this word does not belong in a scientific journal of this caliber.

10) I thank the authors for sharing the materials, data, and analysis scripts in the Open Science Framework (OSF) page. In the open access SPSS file, the number of identified Ss is 293, not 243. Details of how Ss were treated with respect to recruitment vs. data analysis are missing in the current manuscript.

11) Lines 168-169: "Third, as the T system is sexually dimorphic, and given that most of the behavioral literature in animals is concerned with males, we relied on a male sample." There should be a compelling reason why the authors specifically chose men in their sample, especially given that women engage in conspicuous consumption. Saying that they chose men on the basis that most studies chose men weakens their paper.

Related to this point, in lines 182-189, they argue that their research overcomes some limitations in previous studies, two of which used only female samples (Eisenegger et al., 2010; van Honk et al., 2012). Lots of studies now are looking at testosterone and women, e.g., the work of Pranjali Mehta and Oliver Schultheiss, among others.

12) Lines 86-87: Ruling out alternative explanations here seems both hurried and premature. This should come with the presentation/discussion of findings.

13) I would let the sample size speak for itself, mostly because one can never be sure of how many Ss are included in all similar studies taking place right now.

14) Lines 109-116 should be revisited and examples rethought. First, power and status must be better differentiated, especially since these directly relate to the DVs. Second, while the English monarch leaves most of the decision making to the British parliament, it is incorrect to suggest that she has no power. After all, millions of tourists visit Buckingham Palace every year so she must have some influence. Also, from a more technical point of view, she not only owns 24 Sussex Drive, the home of the Canadian Prime Minister (resource control), but she can also legitimately declare war on another country (i.e., the Royal Prerogative).

15) How does the research design assure "ecological validity" (line 162)? A related question could be: how do the experimental findings help inform consumer attitudes among men in the marketplace?

16) Lines 191-202: The last two paragraphs, at least to me, constitute major conceptual leaps or hyperbolic statements. For example, in the first paragraph, the authors say, "Our findings may have further implications for a better understanding of T's role in social behaviors of non-human species." I am completely failing to see how giving exogenous testosterone to human males and having them express their attitudes vis-à-vis brands leads us to better understand biosocial interactions of animals (where status and power are rarely decoupled). How do the results here specifically suggest that testosterone promotes aggressive behavior in animals?

An equally confusing paragraph (and the confusing quote from the LVMH Head) is how the research here helps to explain the sustained commercial success of luxury goods in general. Are men who are

taking testosterone supplements nowadays purchasing more luxurious goods? I am failing to connect the findings described here with the much larger implication.

17) Lines 226-227: When providing a rationale for conducting the battery of studies, "because it looked favorably upon by IRBs" is rarely a good response. The authors ought to rethink this statement. Were there other behavioral exercises carried out in between the collected saliva samples which were not mentioned here? Also, in line 257, I am not sure that explaining what a double-blind study means is required with the readership of this journal.

18) How much saliva, in milliliters, was collected in each instance in order to be able to measure over one dozen steroids?

19) What were participants told prior to engaging in the study (purpose of the research as stated in the script)? Apart from adding one brand comparison pair (GAP vs. H&M; both low in status), what other measures were taken to ensure that the Ss did not guess the hypotheses?

20) Both intra- and inter-assay coefficients of variation (for the samples tested) are provided when reporting testosterone measures. Could these be provided?

21) There should be some justification for using control variables in any experimental study. Here, the results confirmed hypotheses irrespective of controls. Did the authors then add the excluded Ss - i.e., those on the basis of age and mood - and rerun the analyses?

22) I appreciate the authors' transparency in lines 648-661 regarding the local testosterone spread, and the remedies enacted to prevent its further spread. More clarity is needed, however, to evaluate the extent to which this is problematic.

Reviewer #3 (Remarks to the Author):

This is an interesting paper that provides support for the claim that testosterone increases men's preferences for status goods. A reliable and clear assessment of the behavioural effects of testosterone in humans is still lacking so that this studies like this one are welcome.

It is a placebo-controlled testosterone administration study with an unusually large sample in which young healthy men were administered 10mg of a testosterone or placebo gel. The preference for status goods was measured by a questionnaire in which participants indicated their preference for a high status/power good relative to a low status/power good. In a second task the authors distinguished status preferences from power preferences. Participants expressed their attitude towards various goods that were either described as high-status, high-power or high-quality goods. Testosterone improved attitudes to goods described as high-status but not to goods described as high-power or high-quality. The data shows a significant testosterone related increase in the hypothetical willingness to pay for status goods and in intentions to buy these goods but the effect is only significant at the 10% level with a two-sided test. In my view the authors should and – given the previous evidence on the role of testosterone on preferences for status – can justify the use of a one-sided test. Based on the above mentioned observations, the authors conclude that testosterone increases status preferences but not power preferences.

This is an interesting and important study with some clear results. The experimental and statistical methods are sound and described in detail. The large number of participants is also a great plus of this study. The authors extend the testosterone-status literature in an important dimension – to status-

driven consumption goods.

Here are a few suggestions how the authors should further improve their study:

1) The most important point concerns task 2, the authors claim that the effects are driven by status enhancement, not by power enhancement nor by quality considerations. However, this claim is not yet supported by the evidence provided. They report whether 'status' and 'power' enhancement differ from the baseline condition, i.e. when subjects face the quality good. From this they can conclude that status enhancement was stronger than quality consideration, but they cannot make claims regarding the difference between status and power enhancement. For such a claim they would need to run a further model with power as a baseline or they need to study the interaction effect. The authors should do this in their revision of the paper.

2) The authors fail to cite original and recent articles that link testosterone and status/power preferences. Among others, I would expect the authors to cite and discuss among others the various articles by Schultheiss, van Honk or Eisenegger on the link between testosterone and status/power.

3) The authors claim in the last paragraph that their results have implications for the consumption of luxury goods. Given that the authors did not use incentivised ratings or purchasing decisions, this conclusion is not yet fully justified in my view. The effects of testosterone on hypothetical purchasing decisions appear rather weak. If testosterone has only weak effects on hypothetical purchasing decisions of luxury goods it is not so clear how it would affect actual purchases.

Response to Reviewer 1

We would like to thank the Reviewer for his/her encouraging feedback and insightful comments, which we have taken to heart in preparing this revision.

For the Reviewer's convenience, below we have copied his/her comments verbatim in **bold typeface**.

1.1 This paper demonstrates that the exogenous administration of testosterone (via a gel) to a group of men increases their preference for status products (across two different experimental tasks). The paper was clearly written, and the methodology and associated data analyses seemed appropriate. To the extent that this causal link has yet to be empirically demonstrated using an exogenous administration of testosterone, it offers a novel empirical insight. That said, the finding fails on a very important epistemological criterion, as originally set forth by Davis (1971). Regrettably, it is unlikely to trigger an emphatic "That's Interesting!" response across the interdisciplinary readership of Nature Communications. Furthermore, the main finding (while novel in a strict empirical sense is not so from a theoretical perspective) does not seem to pass the necessary threshold that would warrant publication. The current paper seems best suited for a flash report (or short report) in a more specialized journal.

We thank the Reviewer for this assessment of our work.

1.2 I should add that there is an important theoretical issue that was not addressed in this paper. Is the exogenous administration of testosterone meant to mimic (in an ecological senses) the situational and temporary increase in testosterone (e.g., as might occur via an increase in one's social status following a job promotion)? A more dispositional (rather than situational) perspective would posit that since men exhibit individual differences in terms of their baseline circulating testosterone levels, those with higher baseline T levels are more likely to prefer status products. Hence, a possible additional study that would increase the contribution of this work would seek to establish whether or not the dispositional T effect is operative.

We thank the Reviewer for raising this important point. The exogenous administration of testosterone in our study was meant to mimic the activational (i.e., situational) increase in testosterone that is typically influenced by context (e.g., an interaction with an attractive potential mate, or winning a competition). In this sense, our work is similar to several high-profile publications that reported behavioral effects of testosterone

administration on economic behavior (van Honk et al. 2016; Eisenegger et al. 2010; van Honk et al. 2012; Dreher et al. 2016; Boksem et al. 2013; Wibrat et al. 2012). In most of these studies, baseline testosterone levels were not measured, and no claims about dispositional effects were made.

We agree, however, with the Reviewer that the question as to whether a dispositional testosterone effect is also operative is interesting, and we are happy to reply that we can address it using our current data, as we collected baseline (pre-treatment) saliva measurements of testosterone and its metabolites (originally collected for the purpose of a manipulation check). Note that these testosterone measurements were obtained using mass spectrometry (the most accurate measurement method), and our sample size is much larger than those used in most previous studies investigating dispositional effects (e.g., Apicella et al. 2011, 2008; Campbell et al. 2010; Stanton et al. 2011; Coates and Herbert 2008).

In the revised manuscript, we now report new analyses that are identical to the ones testing for an effect of testosterone administration, but this time we also included the logged baseline (pre-treatment) measurements of testosterone. Please note that all effects are robust to omission of the treatment effect from the model, but we decided to include it as treatment was shown to influence the task. As a robustness check, we also performed these analyses with baseline levels of androstenedione, a downstream metabolite of testosterone, levels of which correlate with testosterone (in our sample the correlation was 0.44, $p < 0.01$), and which has been suggested (by animal studies) to have potent behavioral effects similar to those of testosterone (Payne 1974; Tsutsui and Ishii 1981).

In task 1, we found that indeed, baseline levels of both testosterone and androstenedione were positively and robustly associated with preferences for the high-status brands (for testosterone: $\beta = 0.13$, 95% CI = [0.042 0.220], $z = 2.867$, $p = 0.004$; for androstenedione: $\beta = 0.301$, 95% CI = [0.084 0.518], $z = 2.708293$, $p = 0.007$). (Note that the treatment effect remains robust when controlling for the baseline levels.) This finding suggests that dispositional effects of testosterone and androstenedione also play a role in this task.

In task 2, however, we were not able to detect a reliable association between testosterone and preferences for status, as we did for the exogenous treatment (the testosterone x status coefficient was positive, but not statistically significant). It is important to keep in mind that although they study the same psychological constructs, the two tasks are not identical. In task 1 we measured preferences for status by contrasting familiar (high vs. low rank) brands, whereas in task 2 we kept the brands

and products constant and manipulated only the description of the products to impute status, power, or quality to the same products. It is also important to keep in mind that the dispositional results are correlational, and do not allow causal inference.

Taken together, our findings are suggestive that a dispositional effect may be operative, but more research is needed to establish its robustness.

Following these analyses, we added the following to the main text:

When describing the results of study 1 (p. 4):

Additional analyses revealed that participants' baseline (pre-treatment) T levels were also associated with greater preference for brands with higher social rank ($\beta = 0.13$, 95% CI = [0.042 0.220], $z = 2.867$, $p = 0.004$, see Table S12), suggesting that the effect of T is both "activational" (i.e., behavior is influenced by a transient change in T levels) and "dispositional" (i.e., behavior is related to baseline T levels, which are relatively stable when measured in the individual at the same time of day).

Footnote: *We found a similar effect for baseline levels of androstenedione ($z = 2.53$, $p < 0.01$), a downstream metabolite of testosterone. Levels of androstenedione correlate with testosterone (in our sample $r = 0.44$), and it has been suggested (by animal studies) to have potent behavioral effects that are similar to those of testosterone (for further details, see SI).*

When describing the results of study 2 (pp. 5–6):

As in the analysis of task 1, we performed additional analyses exploring the association between participants' baseline (pre-treatment) T and their attitudes toward the goods. However, we did not detect a reliable baseline T \times status interaction, though the coefficient was positive in sign (see Table S13).

In the discussion (p. 7):

Moreover, given that we found a reliable association between baseline T levels and consumer preferences only in the first task, and this is the first study testing this relationship, more research is needed to establish robustness and variation of a dispositional effect of T on consumer preferences.

1.3 Finally, several key references are missing. For example, when discussing Zahavian signaling in a consumer setting, it is imperative that the authors cite Saad (2007), as he offered a detailed analysis of this exact mechanism. Furthermore, when discussing the digit ratio, the authors should cite the

numerous recent papers that have been published linking the digit ratio to consumer-related phenomena (cf. Nepomuceno et al., 2016a, 2016b). Others papers worth citing including Nelissen and Meijers (2011) and Stanton (2017).

Thank you for these suggestions. We agree that these works are highly relevant.

We added references to Saad (2007) and Nelissen and Meijers (2011) when introducing the concept of conspicuous consumption.

We added a reference to the Stanton (2017) review when introducing the relationship between testosterone and decision-making (together with Eisenegger et al. 2011).

We were initially hesitant to include reference to the 2D:4D studies, as the use of the 2D:4D as a proxy for prenatal testosterone has been challenged by several researchers on methodological grounds (e.g., Apicella et al. 2016; Berenbaum et al. 2009), and because evidence from recent large-scale meta-analyses concluded that the correlations between the 2D:4D and traits that are thought to be related to prenatal testosterone exposure were meager in magnitude—and would require samples of thousands of participants to detect (e.g., aggression (Pratt, Turanovic, and Cullen 2016; Turanovic, Pratt, and Piquero 2017) , sexual orientation (Voracek et al. 2011)) . Moreover, our own results reported in the paper show that 2D:4D does not reliably correlate with consumer preferences in our sample of young men (which is larger than previous investigations of the subject). To our view, establishing the robustness of these 2D:4D findings would require more conclusive results from further replications and larger samples.

In the new version of the manuscript we included a footnote referring to these works, while also referring the reader to a study highlighting the limitation of this line of research (p. 3):

Footnote: Other studies reported intriguing correlations between the 2D:4D digit ratio, a candidate proxy of prenatal testosterone exposure (though see ³⁸), and behavioral measures of courtship-related consumption,^{39,40} although this relation is not evident in our data.

References:

Davis, M. S. (1971). That's interesting! Towards a phenomenology of sociology and a sociology of phenomenology. *Philosophy of Social Science*, 1, 309–344.

Nepomuceno, M., Saad, G., Stenstrom, E., Mendenhall, Z., & Iglesias, F. (2016a). Testosterone & gift giving: Mating confidence moderates the association between digit ratios (2D4D and rel2) and erotic gift giving. *Personality and Individual Differences*, 91, 27–30.

Nepomuceno, M., Saad, G., Stenstrom, E., Mendenhall, Z., & Iglesias, F. (2016b). Testosterone at your fingertips: Digit ratios (2D:4D and rel2) as predictors of courtship-related consumption intended to acquire and retain mates. *Journal of Consumer Psychology*, 26, 231–244.

Nelissen, R. M. A., and Meijers, M. H. C. (2011). Social benefits of luxury brands as costly signals of wealth and status. *Evolution and Human Behavior*, 32, 343–355.

Saad, G. (2007). *The Evolutionary Bases of Consumption*. Mahwah, NJ: Lawrence Erlbaum.

Stanton, S. J. (2017). The role of testosterone and estrogen in consumer behavior and social & economic decision making: A review. *Hormones and Behavior*, 92, 155-163.

References

- Apicella, Coren L., Anna Dreber, Benjamin Campbell, Peter B. Gray, Moshe Hoffman, and Anthony C. Little. 2008. "Testosterone and Financial Risk Preferences." *Evolution and Human Behavior: Official Journal of the Human Behavior and Evolution Society* 29(6):384–90.
- Apicella, Coren L., Anna Dreber, Peter B. Gray, Moshe Hoffman, Anthony C. Little, and Benjamin C. Campbell. 2011. "Androgens and Competitiveness in Men." *Journal of Neuroscience, Psychology, and Economics* 4(1):54.
- Apicella, Coren L., Victoria A. Tobolsky, Frank W. Marlowe, and Kathleen W. Miller. 2016. "Hadza Hunter-Gatherer Men Do Not Have More Masculine Digit Ratios (2D:4D)." *American Journal of Physical Anthropology* 159(2):223–32.
- Berenbaum, Sheri A., Kristina Korman Bryk, Nicole Nowak, Charmian A. Quigley, and Scott Moffat. 2009. "Fingers as a Marker of Prenatal Androgen Exposure." *Endocrinology* 150(11):5119–24.
- Boksem, Maarten A. S., Pranjal H. Mehta, Bram Van den Bergh, Veerle van Son, Stefan T. Trautmann, Karin Roelofs, Ale Smidts, and Alan G. Sanfey. 2013. "Testosterone Inhibits Trust but Promotes Reciprocity." *Psychological Science* 24(11):2306–14.
- Campbell, Benjamin C., Anna Dreber, Coren L. Apicella, Dan T. A. Eisenberg, Peter B. Gray, Anthony C. Little, Justin R. Garcia, Richard S. Zamore, and J. Koji Lum. 2010. "Testosterone Exposure, Dopaminergic Reward, and Sensation-Seeking in Young Men." *Physiology & Behavior* 99(4):451–56.
- Coates, J. M., and J. Herbert. 2008. "Endogenous Steroids and Financial Risk Taking on a London Trading Floor." *Proceedings of the National Academy of Sciences of*

- the United States of America* 105(16):6167–72.
- Dreher, Jean-Claude, Simon Dunne, Agnieszka Pazderska, Thomas Frodl, John J. Nolan, and John P. O'Doherty. 2016. "Testosterone Causes Both Prosocial and Antisocial Status-Enhancing Behaviors in Human Males." *Proceedings of the National Academy of Sciences* 113(41):11633–38.
- Eisenegger, C., M. Naef, R. Snozzi, M. Heinrichs, and E. Fehr. 2010. "Prejudice and Truth about the Effect of Testosterone on Human Bargaining Behaviour." *Nature* 463(7279):356–59.
- Honk, Jack van, Estrella R. Montoya, Peter A. Bos, Mark van Vugt, and David Terburg. 2012. "New Evidence on Testosterone and Cooperation." *Nature* 485(7399):E4–5; discussion E5–6.
- Honk, Jack van, Geert-Jan Will, David Terburg, Werner Raub, Christoph Eisenegger, and Vincent Buskens. 2016. "Effects of Testosterone Administration on Strategic Gambling in Poker Play." *Scientific Reports* 6(January):18096.
- Payne, A. P. 1974. "A Comparison of the Effects of Androstenedione, Dihydrotestosterone and Testosterone Propionate on Aggression in the Castrated Male Golden Hamster." *Physiology & Behavior* 13(1):21–26.
- Pratt, Travis C., Jillian J. Turanovic, and Francis T. Cullen. 2016. "Revisiting the Criminological Consequences of Exposure to Fetal Testosterone: A Meta-Analysis of the 2D:4D Digit Ratio." *Criminology: An Interdisciplinary Journal* 54(4):587–620.
- Saad, Gad, and John G. Vongas. 2009. "The Effect of Conspicuous Consumption on Men's Testosterone Levels." *Organizational Behavior and Human Decision Processes* 110(2):80–92.
- Stanton, Steven J., O'Dhaniel A. Mullette-Gillman, R. Edward McLaurin, Cynthia M. Kuhn, Kevin S. LaBar, Michael L. Platt, and Scott A. Huettel. 2011. "Low- and High-Testosterone Individuals Exhibit Decreased Aversion to Economic Risk." *Psychological Science* 22(4):447–53.
- Tsutsui, K., and S. Ishii. 1981. "Effects of Sex Steroids on Aggressive Behavior of Adult Male Japanese Quail." *General and Comparative Endocrinology* 44(4), 480–86.
- Turanovic, Jillian J., Travis C. Pratt, and Alex R. Piquero. 2017. "Exposure to Fetal Testosterone, Aggression, and Violent Behavior: A Meta-Analysis of the 2D:4D Digit Ratio." *Aggression and Violent Behavior* 33(March):51–61.
- Voracek, Martin, Jakob Pietschnig, Ingo W. Nader, and Stefan Stieger. 2011. "Digit Ratio (2D:4D) and Sex-Role Orientation: Further Evidence and Meta-Analysis." *Personality and Individual Differences* 51(4):417–22.
- Wibral, Matthias, Thomas Dohmen, Dietrich Klingmüller, Bernd Weber, and Armin Falk. 2012. "Testosterone Administration Reduces Lying in Men." *PloS One* 7(10):e46774.

Response to Reviewer 2

We would like to thank the Reviewer for his/her encouraging feedback and insightful comments, which we have taken to heart in preparing this revision.

For the Reviewer's convenience, below we have copied his/her comments in **bold typeface**.

In this article, the authors seek to provide causal evidence that exogenous testosterone administration in a sample of men increases their preference for status brands, as well as favorable attitudes toward goods that are indicative of social status compared to those suggestive of either power or quality.

I commend the authors for having undertaken an intricate and laborious experiment lasting several hours. It was fun to read. However, I point to some important limitations. They are included below, in no particular order:

2.1 The authors mention that we don't know much about the biological underpinnings of status-driven consumption. First, why is this important? This question remains unanswered throughout the paper.

Thank you for this important point. We agree that we did not sufficiently impart the importance and implications of understanding the biological basis of status-driven consumption. In response to this comment, we added two paragraphs to the manuscript.

In the revised introduction, we added a paragraph highlighting why understanding the drivers of status-driven consumption is important (pp. 2–3):

Understanding the drivers of costly signaling through positional consumption is important because this behavior is, by definition, wasteful—in the sense that less expensive goods could have the same functional value as more expensive status goods (think about cars and houses, as examples). Status competition therefore creates inefficiencies. Spending resources to elevate perceived status might, for example, perpetuate poverty by reducing self-investment in health and education among the poor, who appear to spend disproportionately more on status signals and thus substitute status signaled through consumption for long-run wealth accumulation.^{17,18,19} While recent work has explored the socio-psychological antecedents of status-driven consumption,²⁰⁻²² little is known about its biological basis, via genes, hormones, or brain activity.

In the revised discussion, we added a paragraph highlighting how our finding of a biological factor underlying conspicuous consumption may lead to novel predictions about the contexts in which conspicuous consumption might arise (p. 6):

Our findings may be useful for generating new hypotheses regarding contexts in which conspicuous consumption occurs.^{17,18,19} Men experience situational elevation in T during and following sporting events, in the presence of attractive mates, and following meaningful life events such as graduation and divorce.^{29,46,47} Our results suggest that in such contexts, male consumers might be more likely to engage in positional consumption, and might find status-related brand communications more appealing. We hope that our findings will guide further research exploring how contextual variations in real-world settings that change T levels (e.g., sporting events, changes in marital status) affect status-enhancing economic preferences.

Second, when they discuss the connection between status and male testosterone, it is mostly in nonhumans. Related to this point is what Sapolsky (1997) describes as a limitation in our understanding of behavioral endocrinology, namely that we often (and mistakenly) believe that our testosterone is related to behavior; instead, we should ask whether the opposite is at play. As such, I think that the authors should be open to the possibility of reverse causality here when discussing their limitations.

Sapolsky, R.M. (1997). The trouble with testosterone. New York, NY: Scribner.

We thank the reviewer for making this point.

It is important to note that in this study we *randomly assigned* participants to experimental treatments of testosterone or placebo, and this methodology permits causal inference in the direction that we proposed. Further, we designed the experiment and its individual components to omit feedback about task performance, thereby eliminating the effects of behavior on testosterone and closing this potential bidirectional, confounding channel. Despite this careful design choice, there is the potential for individual variation in *response* to the tasks, yet this should not have an influence on the estimated treatment effect in a random assignment paradigm.

Having said that, we certainly agree that causality in the other direction also exists. (We also noted in the previous version that such a reverse effect has been demonstrated empirically, by Saad and Vongas 2009.)

To clarify that the relationship between testosterone and behavior is dynamic, and to call for further research investigating this interplay, we added the following to our discussion of limitations (p. 7):

Third, the causal connection between T and status appears to go in both directions. We find that changes in T causally promote positive attitudes toward status-enhancing

goods. Other studies have shown that consumption of such goods also causes changes in T levels.³⁰ This bi-directional relationship suggests a testosterone-mediated process in which consumption of status products might be self-reinforcing,⁸ and calls for further investigations of the dynamics of how testosterone and consumption influence each other over time.

2.2 In lines 44-49, the authors could have captured Veblen's basic idea more effectively. In his treatise, he criticized contemporary economists for their use of classical market theories to explain a growing and puzzling phenomenon. Using other disparate disciplines (e.g., anthropology), he described that, as societies shifted from hunter gathering to agriculture and more industrialized pursuits, accumulated wealth replaced aggression as a means for men to signal their standing in society. On a related point, Veblen's book was published in 1899; the reference included here should not contain Martha Banta's name.

Thank you for these important points. We have made efforts to incorporate them when introducing the concept of conspicuous consumption (p. 2):

*How do people achieve higher status? In early human societies, displays of hunting skills and physical aggression were the primary means used to promote one's standing in society. In contemporary settings, however, hunting and aggression have been replaced by different strategies, such as displays of culturally valued skills and behaviors (e.g., the obtention of academic degrees). Another prevalent route to greater status rests on the display of accumulated wealth, through conspicuous economic consumption.^{10,11} This idea was introduced by Thorstein Veblen's seminal work, *The Theory of the Leisure Class*,¹² which describes how wasteful expenditures on positional goods, which display one's apparent resources to others, shape the social strata over time.⁸ Such goods are particularly effective signals of status because they separate the "haves" from the "have nots" through economic (e.g., high price) or physical (e.g., restricted access for private club members) barriers. Although Veblen's insights were overlooked by classical market theories, modern economic theories have described how a balance of prices and goods sustains conspicuous consumption as a costly signal.^{13,14} Indeed, goods that wealthier individuals are more likely to buy (hereafter, "positional goods") also tend to be more visible to others than other goods that are more affordable and thus accessible to everyone.^{15, 16}*

2.3 Line 68: "animals and humans..." Humans are animals (Kingdom Animalia)!

Thank you for pointing this out. We replaced this expression with "non-human and human species" (p. 3).

2.4 Lines 71-72: “subsequent research has proposed indirect evidence that T does not increase aggression per se...” One recent study, in fact, found direct evidence that testosterone increases did not lead to various forms of aggression. Here is the reference:

Vongas, J.G., & Al Hajj, R. (2017). The effects of competition and implicit power motive on men’s testosterone, emotion recognition, and aggression. *Hormones & Behavior*, 92, 57-71.

We thank the Reviewer for directing us to this highly relevant paper that was published recently. In response, we removed the words “indirect evidence” and added a citation referring the readers to this work (p. 3).

2.5 In line 77, the authors should note that this paper (Saad & Vongas, 2009) does not report any empirical tests of mediation regarding audience effects (i.e., “visibility of the consuming act”). This should be fixed because it is misleading in the current form.

Thank you for reading carefully. We agree, and we removed the claim about visibility from the paper.

2.6 Apart from what previous studies have found, what is the theoretical basis for formulating the hypothesis in the current manuscript (lines 79-80)? Papers that are presented by the authors have been guided by a theoretical framework (e.g., Zahavi’s handicap principle or costly signaling theory) so readers might be left guessing as to what is driving the hypotheses here.

We thank the reviewer for this important point. In response, we added the following text when presenting our hypothesis (p. 3):

Building on Veblen’s theory of conspicuous consumption, as well as the evidence that a situational increase in T leads to rank-promoting behaviors in non-humans and humans, we hypothesized that elevated T levels would cause men to exhibit stronger preference toward goods that promote their social rank.

2.7 Subject effect: Given that it was not possible to control what Ss did in between getting the testosterone gel application and returning to the lab, and apart from guidelines to avoid contact with females, what other measures did the authors use to protect the study’s internal validity? Was there a manipulation

check at the end of the experiment, prior to debriefing, that gauged the extent to which Ss might have guessed the hypotheses being tested here (and to exclude participants on this basis)?

Thank you for raising this important issue. Our manipulation checks focused on several measures. One of them was possible changes in mood due to the treatment since mood has been shown to impact economic preferences and consumer behavior (e.g., Lerner et al. 2004). We also asked the participants to guess the treatment they had received, another factor that was previously shown to influence economic behavior (Eisenegger et al. 2010) and found no differences between the two treatment groups (see Table S1). As direct manipulation check for the impact of T administration between groups we collected four separate saliva samples throughout the experiment (analyzed for 14 hormones by LC-MS/MS) to monitor androgenic and other hormonal changes with the prediction that participants in the treatment but not in the placebo group should have significantly higher levels of T and its metabolites, which was confirmed in our analysis (see Table S3). Furthermore, we found no differences between the two treatment groups in the levels of all other hormonal measurements, such as cortisol.

However, we did not ask participants to guess the behavioral hypotheses which would have been another measure to check for potential demand effects as the reviewer suggests. As the treatment was randomized and participants could not guess what treatment they had received, it is unlikely though that our results for treatment influence on the task are driven by potential demand effects. We agree nevertheless that the measures we implemented detailed above do not entirely rule out any possible channel of influence (including the possibility of an indirect effect mediated by participants' behavior during the loading period), and acknowledge this as a limitation in the current version of the manuscript (p. 7):

Furthermore, while the effects reported here were robust to controlling for mood, beliefs about treatment and the levels of other hormones that might have influenced behavior, we cannot entirely rule out that the effects of T on economic behavior were generated by other indirect mechanisms. For example, we could not fully control participants' behaviors during the drug loading period (i.e., the time interval between when the T gel was applied and when they returned to the lab).

2.8 Related to the above point #7: How much were Ss paid for participating?

Participants earned on average \$68.12 USD (SD = \$17.36) overall. We added the following to the materials and methods section (p. 8):

On average, participants earned \$68.12 USD (SD = \$17.36) for participating in the experiment, where payout varied as a function of their performance in some of the other tasks.

2.9 Line 147: The authors should remove “exciting.” While I agree that future research opportunities are exciting, I feel this word does not belong in a scientific journal of this caliber.

We agree and removed the word “exciting”.

2.10 I thank the authors for sharing the materials, data, and analysis scripts in the Open Science Framework (OSF) page. In the open access SPSS file, the number of identified Ss is 293, not 243. Details of how Ss were treated with respect to recruitment vs. data analysis are missing in the current manuscript.

We thank the reviewer for reading carefully and going over the materials. All of the participants (N = 243) are included in the analysis, as described in the paper. As the reviewer correctly pointed out, the participant IDs indeed go up to 293; this is because we did not use all of the numbers between 1 and 293 when assigning IDs to the participants. Specifically, (1) we started the ID numbering from 11 (the numbers between 1 and 10 were used by research assistants, who completed the task as pilot participants and did not receive a pharmacological treatment), and (2) we deliberately left several “empty” numbers between sessions (e.g., the numbers between 23 and 26 were not used).

2.11 Lines 168-169: “Third, as the T system is sexually dimorphic, and given that most of the behavioral literature in animals is concerned with males, we relied on a male sample.” There should be a compelling reason why the authors specifically chose men in their sample, especially given that women engage in conspicuous consumption. Saying that they chose men on the basis that most studies chose men weakens their paper.

Related to this point, in lines 182-189, they argue that their research overcomes some limitations in previous studies, two of which used only female samples (Eisenegger et al., 2010; van Honk et al., 2012). Lots of studies now are looking at testosterone and women, e.g., the work of Pranjali Mehta and Oliver Schultheiss, among others.

We thank the Reviewer for pushing us on this important issue. Our decision to focus on male participants was based on several theoretical and methodological reasons.

First, we reiterate that most of the behavioral literature in non-human animals is concerned with males, and our theoretical predictions were inspired by this literature.

Second, as the testosterone system is sexually dimorphic, studying only one sex in an experiment is common practice in the field. (As far as we know, all previous T administration studies used participants who were all of the same sex.)

Third, we agree with the reviewer that women, too, engage in positional consumption. However, from a practical standpoint, neither the stimuli nor the treatment in our study is identically applicable to women, because (a) the brands, products, and marketing messages that would manipulate status motives differ between women and men, and therefore testing our prediction in women would require changing the behavioral tasks to reflect these sex differences, (b) T-gel is not FDA-approved for women, and administering T to women would require dosage adjustment and extraordinary measures to obtain ethics approval, and (c) the most prominent bio-social theories of positional consumption in women are concerned with fertility phases during the menstrual cycle and the hormones controlling them, estrogen and progesterone (Lens et al. 2012; Durante et al. 2011; Saad and Stenstrom 2012; Durante et al. 2014; Eisenbruch, Simmons, and Roney 2015). While it is likely that T plays a role too, it is not the primary candidate endocrinological mechanism, and studying it in women would require additional features that complicate the experimental design further (because menstrual cycle phase induces an additional source of variation to the data, and must be carefully controlled for).

Of course, we could have designed different stimuli, used a different administration method, and carefully monitored the menstrual cycles of female participants. However, this would have come with a large cost, and as in the current study we maximized sample size given a budget constraint, this would imply reducing the number of participants in the male study by at least a half. As small-sample studies are not capable of reliably detecting small effects and are associated with higher type I and type II errors (Gelman and Carlin 2014; Button et al. 2013; Walum, Waldman, and Young 2016), we decided to invest our resources in maximizing statistical power by studying the sex for which there is a stronger theoretical justification.

On a final note, we are aware that recent studies have investigated T levels in women. However, this line of research is still in its infancy and suffers from serious methodological limitations—including systematic upwards biases in the measurements of women's T levels by the most commonly used kits, as noted by a recent important paper coming from Pranjali Mehta's lab (Welker et al. 2016). In line with this finding, some of the previously published studies reported T values that are several times higher than what one should expect to see in women (e.g., Sapienza, Zingales, and Maestripieri 2009). Furthermore, the only relevant study that we are aware of, which reported an association between testosterone and a relevant behavioral measure—the

choice of attractive makeup—has been retracted on methodological grounds (Fisher et al. 2015).

In response to this comment, we added the following to the discussion (p. 7):

Fourth, because the T system is sexually dimorphic, and given that most of the behavioral literature in animals is concerned with males, we relied on an all-male sample (the use of same-sex participants is a common practice in the literature). It is important to note, however, that women also engage in positional consumption, and preliminary evidence suggests that biological factors (including hormones that relate to the menstrual cycle) are involved.⁵⁰ As there is evidence that T promotes status-related behaviors in females,^{24,35,36} further research should explore whether the effects of T on consumer preferences are generalizable to females, while taking into account that which brands and goods are status-enhancing is likely to differ across sexes.⁵¹

2.12 Lines 86-87: Ruling out alternative explanations here seems both hurried and premature. This should come with the presentation/discussion of findings.

We agree with the Reviewer. We removed the sentence that she/he is referring to, and kept the note that we control for the factors mentioned (mood, treatment expectancy, demographics) in the results section.

2.13 I would let the sample size speak for itself, mostly because one can never be sure of how many Ss are included in all similar studies taking place right now.

We agree, and removed the comment about our study being the “largest to date.”

2.14 Lines 109-116 should be revisited and examples rethought. First, power and status must be better differentiated, especially since these directly relate to the DVs. Second, while the English monarch leaves most of the decision making to the British parliament, it is incorrect to suggest that she has no power. After all, millions of tourists visit Buckingham Palace every year so she must have some influence. Also, from a more technical point of view, she not only owns 24 Sussex Drive, the home of the Canadian Prime Minister (resource control), but she can also legitimately declare war on another country (i.e., the Royal Prerogative).

We thank the Reviewer for making these points. Following his/her comments, we rewrote the paragraph to better differentiate status and power, and also changed the example he/she is referring to (pp. 4–5):

The second task investigated T's effect on two distinct ways individuals can use consumption to climb the social ladder: increasing their power (defined as feelings of control over valuable resources) and status (the prestige, respect, and admiration an individual has in the eyes of others).^{8,9} Although power and status are inextricably intertwined in most animal social groups, the two can be decoupled in humans. For example, a political adviser unknown to the public can have significant control over important decisions without receiving social recognition (high power, low status); conversely, a well-known academic may enjoy high status and be respected by the public, but have little power over policy decisions regarding her research findings.⁸ Thus, human studies provide a unique opportunity to disentangle the extent to which T affects power- versus status-enhancement motives.

2.15 How does the research design assure “ecological validity” (line 162)? A related question could be: how do the experimental findings help inform consumer attitudes among men in the marketplace?

Thank you for pointing this out. In line 162 we refer to the ecological validity of the treatment (not the entire study). We have made this clearer in the new version (p. 7):

“... to assure the ecological validity of our pharmacological treatment, we administered T using a widely prescribed gel at a typical daily dose (100 mg) that leads to elevated serum T levels within the normal male physiological range.”

With respect to the question of how the experimental findings help inform consumer attitudes among men in the marketplace (which is also relevant for the Reviewer's comment 2.1), we added the following to the discussion (p. 6):

Our findings may be useful for generating new hypotheses regarding contexts in which conspicuous consumption occurs.¹⁷⁻¹⁹ Men experience situational elevation in T during and following sporting events, in the presence of attractive mates, and following meaningful life events such as graduation and divorce.^{29,46,47} Our results suggest that in such contexts, male consumers might be more likely to engage in conspicuous consumption, and might find status-related marketing messages more appealing. We hope that our findings will guide further research exploring how contextual variations in real-world settings that change T levels (e.g., sporting events, changes in marital status) affect status-enhancing economic preferences.

2.16 Lines 191-202: The last two paragraphs, at least to me, constitute major conceptual leaps or hyperbolic statements. For example, in the first paragraph, the authors say, “Our findings may have further implications for a better

understanding of T's role in social behaviors of non-human species." I am completely failing to see how giving exogenous testosterone to human males and having them express their attitudes vis-à-vis brands leads us to better understand biosocial interactions of animals (where status and power are rarely decoupled). How do the results here specifically suggest that testosterone promotes aggressive behavior in animals?

An equally confusing paragraph (and the confusing quote from the LVMH Head) is how the research here helps to explain the sustained commercial success of luxury goods in general. Are men who are taking testosterone supplements nowadays purchasing more luxurious goods? I am failing to connect the findings described here with the much larger implication.

We agree with the Reviewer and removed these paragraphs from the paper.

2.17 Lines 226-227: When providing a rationale for conducting the battery of studies, "because it looked favorably upon by IRBs" is rarely a good response. The authors ought to rethink this statement. Were there other behavioral exercises carried out in between the collected saliva samples which were not mentioned here? Also, in line 257, I am not sure that explaining what a double-blind study means is required with the readership of this journal.

Thank you for pointing this issue out. We removed the phrase about IRBs; the current version reads: "*The rationale for conducting a battery of tasks is maximizing the knowledge gained from each human participant undergoing a pharmacological manipulation, a practice that is standard*" (p. 8).

The tasks reported here were the first in the battery and took place immediately after first post-test saliva sample; the other behavioral tasks took place later in the day and therefore have no influence. The current version reads: "*The two tasks reported in the paper were focal and therefore were conducted at the outset, immediately after participants' arrival at the lab in the afternoon and the first post-treatment saliva sample*" (p. 8).

We removed the elaboration on "double-blind."

2.18 How much saliva, in milliliters, was collected in each instance in order to be able to measure over one dozen steroids?

We used LC-MS/MS, which requires minimally about 1 ml of saliva, and we got much more than that from most participants. The saliva was extracted and subjected to a well-validated LC-MS/MS method that monitors up to 15 different steroids and steroid metabolites in the same run.

2.19 What were participants told prior to engaging in the study (purpose of the research as stated in the script)? Apart from adding one brand comparison pair (GAP vs. H&M; both low in status), what other measures were taken to ensure that the Ss did not guess the hypotheses?

We are sorry to be unclear about this important point. We instructed the participants about the goal of the study as follows. Note that we now uploaded all documents mentioned here on the project's Open Science Framework (OSF) page: https://osf.io/jgmnx/?view_only=bc952f56ddd14e27ba2aa53a9fb285c4

1) Recruiting announcement:

“Healthy males 18-55 years of age are invited to participate in a study investigating the effect of the hormone Testosterone on economic decision-making. ... “

2) Consent form:

“This study is designed to investigate biological influences on how you make decisions. The researchers will give you a substance to rub on your shoulders and arms before you make decisions in an asset-trading environment. This substance will either be testosterone (a hormone your body naturally produces that is associated with masculinization during adolescence), or an inert (inactive) substance. Neither you nor the experimenters will know which substance you are getting. ... ”

3) Tasks instructions:

Task 1:

“You will be shown pairs of clothing brands.
For each pair, please indicate your preferred brand, on a scale between 1 and 10.”

Task2:

“You will now participate in a study that aims to understand your preference for different products and advertisements.

In the survey, you will be shown different product ads before being asked questions about them.”

To further clarify, in the debriefing phase, we did not ask participants to guess the hypotheses, but as the treatment was randomized and as participants could not guess

what was the treatment that they had received (as shown by their replies to the question about the belief about the treatment, see Table S1), it is unlikely that our results are biased by potential demand effects. We hope these further details address the reviewers concern.

2.20 Both intra- and inter-assay coefficients of variation (for the samples tested) are provided when reporting testosterone measures. Could these be provided?

Thank you for reading carefully and pointing this out. We have now added the average intra- and inter-assay COVs to Table S2. Please note that the “precision” measure (that was already at the table in the previous version) denotes the range of the inter-assay coefficients of variation over the concentration range measured, where the higher COV would reflect the COV at or near the detection limit (LLOQ).

2.21 There should be some justification for using control variables in any experimental study. Here, the results confirmed hypotheses irrespective of controls. Did the authors then add the excluded Ss - i.e., those on the basis of age and mood - and rerun the analyses?

Thank you for pointing out this lack of clarification. The main analyses are without controls (task 1: model A1 in table S7; task 2: model B1 in table S8), and they include all of the participants in the study (none were excluded). These analyses, therefore, contain more observations than the analyses that include additional control variables (where the participants who did not complete the questionnaires were omitted, as the reviewer correctly notes; see Tables S7 and S8).

Thus, all of the analyses are using all of the participants for whom the relevant data has been collected (we made efforts to better clarify this in the supplementary materials in methods). (Also, please note that when participants were excluded because of missing values, their relative number was small—no more than 7% in any of the analyses.)

2.22 I appreciate the authors’ transparency in lines 648-661 regarding the local testosterone spread, and the remedies enacted to prevent its further spread. More clarity is needed, however, to evaluate the extent to which this is problematic.

As we noted in the supplementary materials of the previous version of the manuscript, we ran several tests to investigate whether the existence of some saliva samples with abnormally high levels pointed to either (a) local spread of T into saliva tubes from

shared surfaces and objects (e.g., pens) or (b) changes in *physiologic* levels from contact with surfaces and objects. We see only the latter as problematic, because it would mean that some of our placebo participants had effectively received T treatment.

Our investigation was based on two tests.

First, the participants who received T showed clear elevation in the levels of its metabolites, DHT and androstenedione. This demonstrates that T was indeed absorbed and metabolized in their bodies. If any of our placebo participants had absorbed T, the levels of its metabolites should have also increased. However, not a single one of the placebo group participants showed abnormally high values of T metabolites in any of the post-treatment measurements. This was also true for the baseline (pre-treatment) samples of *all* participants.

Second, the participants who received testosterone showed a consistent elevation in T levels in all of their post-treatment measurements. This demonstrates that T was elevated during the entire experiment, as expected by this treatment. However, this was not the case in the placebo group, where only five out of 118 participants showed consistently elevated T measurements in all three of the post-treatment saliva samples.

In addition to these findings, we reiterate that previous investigations found that interpersonal T transfer is highly unlikely even with skin-to-skin contact.

Finally, we highlight that our use of saliva samples as a manipulation check for pharmacological testosterone treatment is a substantial methodological improvement in comparison to the previous studies in the literature. Most testosterone administration studies to date—including ones that were published in high-impact outlets such as *Nature* and *PNAS* (Eisenegger et al. 2010; van Honk et al. 2012; Van Honk et al. 2011)—did not collect any biofluids for the sake of a manipulation check, and as far as we know, we are the first to measure salivary testosterone following topical administration. Therefore, we see our detailed report of the local spread issue, our conclusions regarding its implications, and the ways in which we have managed to solve the issue as important methodological contributions of the paper.

References

Button, Katherine S., John P. A. Ioannidis, Claire Mokrysz, Brian A. Nosek, Jonathan Flint, Emma S. J. Robinson, and Marcus R. Munafò. 2013. "Power Failure: Why Small Sample Size Undermines the Reliability of Neuroscience." *Nature Reviews Neuroscience* 14(5):365–76.

- Durante, Kristina M., Vladas Griskevicius, Stephanie M. Cantú, and Jeffrey A. Simpson. 2014. "Money, Status, and the Ovulatory Cycle." *JMR, Journal of Marketing Research* 51(1):27–39.
- Durante, Kristina M., Vladas Griskevicius, Sarah E. Hill, Carin Perilloux, and Norman P. Li. 2011. "Ovulation, Female Competition, and Product Choice: Hormonal Influences on Consumer Behavior." *Journal of Consumer Research* 37(6):921–34.
- Fisher, Claire I., Amanda C. Hahn, Lisa M. DeBruine, and Benedict C. Jones. 2015. "RETRACTED: Women's Preference for Attractive Makeup Tracks Changes in Their Salivary Testosterone." *Psychological Science* 26(12):1958–64.
- Gelman, Andrew, and John Carlin. 2014. "Beyond Power Calculations: Assessing Type S (Sign) and Type M (Magnitude) Errors." *Perspectives on Psychological Science: A Journal of the Association for Psychological Science* 9(6):641–51.
- Lens, Inge, Karolien Driesmans, Mario Pandelaere, and Kim Janssens. 2012. "Would Male Conspicuous Consumption Capture the Female Eye? Menstrual Cycle Effects on Women's Attention to Status Products." *Journal of Experimental Social Psychology* 48(1):346–49.
- Lerner, Jennifer S., Deborah A. Small, and George Loewenstein. "Heart strings and purse strings: Carryover effects of emotions on economic decisions." *Psychological science* 15.5 (2004): 337-341.
- Saad, Gad, and Eric Stenstrom. 2012. "Calories, Beauty, and Ovulation: The Effects of the Menstrual Cycle on Food and Appearance-Related Consumption." *Journal of Consumer Psychology: The Official Journal of the Society for Consumer Psychology* 22(1):102–13.
- Sapientza, Paola, Luigi Zingales, and Dario Maestripieri. 2009. "Gender Differences in Financial Risk Aversion and Career Choices Are Affected by Testosterone." *Proceedings of the National Academy of Sciences of the United States of America* 106(36):15268–73.
- Walum, Hasse, Irwin D. Waldman, and Larry J. Young. 2016. "Statistical and Methodological Considerations for the Interpretation of Intranasal Oxytocin Studies." *Biological Psychiatry* 79(3):251–57.
- Welker, Keith M., Bethany Lassetter, Cassandra M. Brandes, Smrithi Prasad, Dennis R. Koop, and Pranjal H. Mehta. 2016. "A Comparison of Salivary Testosterone Measurement Using Immunoassays and Tandem Mass Spectrometry." *Psychoneuroendocrinology* 71(September):180–88.

Response to Reviewer 3

We would like to thank the Reviewer for his/her encouraging feedback and insightful comments, which we have taken to heart in preparing this revision.

For the Reviewer's convenience, below we have copied his/her comments in **bold typeface**.

3.1 This is an interesting paper that provides support for the claim that testosterone increases men's preferences for status goods. A reliable and clear assessment of the behavioural effects of testosterone in humans is still lacking so that this studies like this one are welcome.

It is a placebo-controlled testosterone administration study with an unusually large sample in which young healthy men were administered 10mg of a testosterone or placebo gel. The preference for status goods was measured by a questionnaire in which participants indicated their preference for a high status/power good relative to a low status/power good. In a second task the authors distinguished status preferences from power preferences. Participants expressed their attitude towards various goods that were either described as high-status, high-power or high-quality goods. Testosterone improved attitudes to goods described as high-status but not to goods described as high-power or high-quality. The data shows a significant testosterone related increase in the hypothetical willingness to pay for status goods and in intentions to buy these goods but the effect is only significant at the 10% level with a two-sided test. In my view the authors should and – given the previous evidence on the role of testosterone on preferences for status – can justify the use of a one-sided test. Based on the above mentioned observations, the authors conclude that testosterone increases status preferences but not power preferences.

This is an interesting and important study with some clear results. The experimental and statistical methods are sound and described in detail. The large number of participants is also a great plus of this study. The authors extend the testosterone-status literature in an important dimension – to status-driven consumption goods.

We thank the Reviewer for his/her supporting comments.

Although we had a clear directional hypothesis, we (a priori) decided to use a conservative two-tailed test because the human testosterone literature is still in its early days, and only a few causal studies had been conducted to date in men (much of the early work was conducted in women). We hope that this work and conservative reporting motivates additional, high-powered studies to elucidate the relationship between T and status preferences.

3.2 Here are a few suggestions how the authors should further improve their study:

1) The most important point concerns task 2, the authors claim that the effects are driven by status enhancement, not by power enhancement nor by quality considerations. However, this claim is not yet supported by the evidence provided. They report whether 'status' and 'power' enhancement differ from the baseline condition, i.e. when subjects face the quality good. From this they can conclude that status enhancement was stronger than quality consideration, but they cannot make claims regarding the difference between status and power enhancement. For such a claim they would need to run a further model with power as a baseline or they need to study the interaction effect. The authors should do this in their revision of the paper.

The Reviewer is correct; this issue was overlooked in the previous analysis. In the current version we added an analysis that focuses on the “power” and “status” conditions, and uses “power” as a baseline (Table S10 in the supplementary materials). We find that all of the effects hold (i.e., preferences for status are greater than preferences for power).

Following this analysis, we added the footnote below to the manuscript (p. 5):

“The results hold when using either quality or power as the baseline category, indicating that T’s positive effect on preferences for status was significantly greater than its effects on preferences for both quality and power (see SI).”

We also added the following to the supplementary results (when reporting extended data analysis and results for task 2) (p. 24):

Finally, we directly tested for differences in attitudes toward power and status ads by applying the same analytical strategy to the power and status ads alone (i.e., excluding quality), when employing the power category as the baseline, and found a significant treatment × status description interaction, demonstrating that participants’ preferences for status-enhancing goods were greater than their preference for status-enhancing goods (Table S11).

2) The authors fail to cite original and recent articles that link testosterone and status/power preferences. Among others, I would expect the authors to cite and discuss among others the various articles by Schultheiss, van Honk or Eisenegger on the link between testosterone and status/power.

Thank you for highlighting these references; we added references to these works to the manuscript.

3) The authors claim in the last paragraph that their results have implications for the consumption of luxury goods. Given that the authors did not use incentivised ratings or purchasing decisions, this conclusion is not yet fully justified in my view. The effects of testosterone on hypothetical purchasing decisions appear rather weak. If testosterone has only weak effects on hypothetical purchasing decisions of luxury goods it is not so clear how it would affect actual purchases.

We agree with the Reviewer and removed the last paragraph from the paper. We have also made efforts to better discuss the lack of incentivized purchasing measures as a limitation that calls for further research (p. 6):

First, the behavioral measures are not actual purchases of goods. Although the pattern of results for our secondary, non-incentivized index of consumer preferences in task 2 was consistent with our primary results on participants' attitudes toward the goods, the effects were smaller in magnitude (statistically significant only at the $p < 0.10$ level, and even weaker when taking into account various controls; see SI)...

... Future research should further investigate T's influence on incentive-compatible economic behaviors promoting social rank, controlling for individual differences as best as possible.

REVIEWERS' COMMENTS:

Reviewer #1 (Remarks to the Author):

I wish to commend the authors on their judicious addressing of the reviewers' comments and suggestions. Well done. A few issues:

1) The authors nicely addressed my feedback regarding the activation versus situational issue. That said, they did not address my "That's Interesting!" issue as enunciated by Davis (1971). Here is how I suggest they do so on a grand epistemological level. In a recent Journal of Marketing Research paper, Saad (2017) explained the epistemological method of evolutionary psychology and how this benefits the work conducted by consumer researchers in particular and behavioral scientists more generally. The authors should add a few sentences that situates their paper within the framework offered by Saad (distinction between proximate and ultimate explanations in consumer research, and the building of nomological networks of cumulative evidence as well as coherent trees of knowledge). The theoretical underpinnings of the authors' current paper is a very nice demonstration of the epistemological method as enunciated by Saad. Now That's Interesting!

Saad, G. (2017). On the method of evolutionary psychology and its applicability to consumer research. Journal of Marketing Research, 54, 464–477.

2) On line 255, the authors cite reference 50. This is fine but they should accordingly cite earlier papers that have examined the effects of the menstrual cycle in consumer settings (e.g., Durante et al., 2011; Saad & Stenstrom, 2012). I noticed that the authors did mention the latter two papers in their reply to reviewer 2 (point 2.11) but yet they do not cite these in the final manuscript. They should.

3) Reviewer 2 (point 2.5) is correct that there was no audience effect in Saad & Vongas (2009) as relating to the high-status car. In this narrower sense, the passage on line 77 of the original paper is incorrect (if referencing solely the effect for the high-status car). However, I had not flagged it in round 1 because there was an audience effect for the low-status car in the following sense (see Saad & Vongas, 2009, p. 85):

"In addition, there was a significant difference between their T levels when they drove the family sedan on the highway and in the downtown area (MFS,H = 2.23, SD = .17 vs. MFS,D = 2.17, SD = .17; t-stat = 3.56, df = 30, p < .001)."

[The subscripts in the above quote were lost when I cut and paste my review onto this online portal.]

That said, these T level changes were not statistically different from the baseline figures (see the second-to-last paragraph on p. 85).

This is why in the abstract the authors had written:

"Additionally, the location of the drive, either a busy downtown area or a semi-deserted highway, partially moderated this response."

This is probably what led the authors of the current paper to write the sentence that they did on line 77 of the originally submitted manuscript. Bottom line: It is incorrect to cite an

audience effect in Saad & Vongas (2009) solely if one is referring to the high-status car. As such, the authors could remove the sentence as suggested by reviewer 2 (point 2.5) or fix it in the manner explained here.

Reviewer #3 (Remarks to the Author):

The authors have responded to all questions and comments of the reviewers in a satisfactory way - also those of reviewer 1 and reviewer 2. I have one remaining question. Does the causal effect of T on preferences for high status brands survive if one controls for baseline T. If YES, then the paper should be published but it needs to be pointed out in the main text that the causal results survive if one controls for baseline T.

Reviewer #1 (Remarks to the Author):

I wish to commend the authors on their judicious addressing of the reviewers' comments and suggestions. Well done. A few issues:

1.1 The authors nicely addressed my feedback regarding the activational versus situational issue. That said, they did not address my "That's Interesting!" issue as enunciated by Davis (1971). Here is how I suggest they do so on a grand epistemological level. In a recent Journal of Marketing Research paper, Saad (2017) explained the epistemological method of evolutionary psychology and how this benefits the work conducted by consumer researchers in particular and behavioral scientists more generally. The authors should add a few sentences that situates their paper within the framework offered by Saad (distinction between proximate and ultimate explanations in consumer research, and the building of nomological networks of cumulative evidence as well as coherent trees of knowledge). The theoretical underpinnings of the authors' current paper is a very nice demonstration of the epistemological method as enunciated by Saad. Now That's Interesting!

Saad, G. (2017). On the method of evolutionary psychology and its applicability to consumer research. Journal of Marketing Research, 54, 464–477.

Thank you for this suggestion. We incorporated it in the first paragraph of the discussion, which now reads as follows:

Taken together, these findings suggest that the consumption of positional goods may stem, at least partly, from biological motives. By adopting an evolutionary perspective, we contribute to a growing body of work in economics uncovering the adaptive function of consumption and complement the increasingly rich nomological network around how status processes govern individuals' thoughts, feelings, and behaviors.

1.2 On line 255, the authors cite reference 50. This is fine but they should accordingly cite earlier papers that have examined the effects of the menstrual cycle in consumer settings (e.g., Durante et al., 2011; Saad & Stenstrom, 2012). I noticed that the authors did mention the latter two papers in their reply to reviewer 2 (point 2.11) but yet they do not cite these in the final manuscript. They should.

We added these two references to the manuscript.

1.3 Reviewer 2 (point 2.5) is correct that there was no audience effect in Saad & Vongas (2009) as relating to the high-status car. In this narrower sense, the passage on line 77 of the original paper is incorrect (if referencing solely the effect for the high-status car). However, I had not flagged it in round 1 because there was an audience effect for the low-status car in the following sense (see Saad & Vongas, 2009, p. 85):

“In addition, there was a significant difference between their T levels when they drove the family sedan on the highway and in the downtown area (MFS,H = 2.23, SD = .17 vs. MFS,D = 2.17, SD = .17; t-stat = 3.56, df = 30, p < .001).”

[The subscripts in the above quote were lost when I cut and paste my review onto this online portal.]

That said, these T level changes were not statistically different from the baseline figures (see the second-to-last paragraph on p. 85).

This is why in the abstract the authors had written:

“Additionally, the location of the drive, either a busy downtown area or a semi-deserted highway, partially moderated this response.”

This is probably what led the authors of the current paper to write the sentence that they did on line 77 of the originally submitted manuscript. Bottom line: It is incorrect to cite an audience effect in Saad & Vongas (2009) solely if one is referring to the high-status car. As such, the authors could remove the sentence as suggested by reviewer 2 (point 2.5) or fix it in the manner explained here.

Thank you for this insightful discussion! We decided to remove the sentence, as suggested by reviewer 2. (This was already done in the previous round.)

Reviewer #3 (Remarks to the Author):

3.1 The authors have responded to all questions and comments of the reviewers in a satisfactory way - also those of reviewer 1 and reviewer 2. I have one remaining question. Does the causal effect of T on preferences for high status

brands survive if one controls for baseline T. If YES, then the paper should be published but it needs to be pointed out in the main text that the causal results survive if one controls for baseline T.

Thank you for pointing this out. Indeed, the effect survives. We have added this to the paper when reporting the baseline T results:

For task 1:

Additional analyses revealed that participants' baseline (pre-treatment) T levels were also associated with greater preference for brands with higher social rank ($\beta = 0.13$, 95% CI = [0.042 0.220], $z = 2.867$, $p = 0.004$; see Supplementary Table 12; the treatment effects were robust to controlling for baseline T).

For task 2:

However, we did not detect a reliable baseline T \times status interaction, though the coefficient was positive in sign (see Supplementary Table 13; the treatment effects were robust to controlling for baseline T).